# Features of Studies on Transition Interventions for Childhood Cancer Survivors: A Scoping Review

**DOI:** 10.3390/cancers16020272

**Published:** 2024-01-08

**Authors:** Jun Ma, Xueling Xiao, Siqi Zhou, Can Gu, Fei Liu, Honghong Wang

**Affiliations:** 1Xiangya School of Nursing, Central South University, Changsha 410013, China; majun_cs@csu.edu.cn (J.M.); xuelingxiao93@hotmail.com (X.X.); zhousiqi@csu.edu.cn (S.Z.); gucan_cs@csu.edu.cn (C.G.); 2Department of Pediatric Hematology Oncology, The Xiangya Hospital, Central South University, Changsha 410008, China

**Keywords:** transition, intervention studies, childhood cancer survivors, scoping review

## Abstract

**Simple Summary:**

Transition interventions can be used to educate and empower childhood cancer survivors (CCSs), preparing them for transition toward assuming adult roles and functions, which can have a sustainable impact on disease prevention and management in long-term survival. The lack of transition interventions is a major barrier for CCSs in transition to adult survivorship care. This review summarized several key aspects of transition interventions, such as delivering knowledge, developing skills for the coordination of care, and addressing psychosocial needs. which may provide strong evidence for facilitating optimal transition and ultimately improve the ability of CCSs to manage their health and health care to maintain their optimal health and well-being.

**Abstract:**

Purpose: in this scoping review, previously reported data were described and synthesized to document transition interventions in CCSs, and the features of intervention components of the current transition studies for CCSs were summarized. Methods: A literature search was conducted in PubMed, Web of Science, EMBASE, PsycINFO, CINAHL, Ovid, and the Cochrane Library following the PRISMA-ScR statement. All original studies (*n* = 9) investigating transition interventions in CCSs were included. Results: The current studies identified essential elements for transition programs, such as delivering knowledge, developing skills for coordination of care, and addressing psychosocial needs. However, the current transition interventions were generally in their infancy, and major deficits were found, including poorly reported intervention components and procedures, a limited number of relevant validated outcomes, and a failure to incorporate conceptual frameworks and international consensus statements. Conclusions: This scoping review mapped current evidence of transition interventions for CCSs and highlighted the paucity of data in this area. More high-quality and well-reported randomized controlled trials are needed for the enrichment and standardization of future transition interventions.

## 1. Introduction

The survival rate for pediatric cancers has steadily increased in recent years [1,2], with >80% of patients now surviving into adulthood [3] and between 39% and 97% within age- and diagnosis-specific groups [4]. Childhood cancer survivors (CCSs) confront a long survivorship phase, and with increasing age, they need to take more responsibility for their health [5]. This is particularly true for survivors who transition through adolescence and enter young adulthood [6]. However, not all CCSs will be well prepared for the transition, and important knowledge deficits exist among CCSs concerning basic details of their diagnosis, previous treatments, and the risk of subsequent late treatment-related effects [7,8], and some CCSs may be more dependent on their parents in terms of knowledge and management skills [9]. In addition, the physical, sexual, psychological, emotional, cognitive, and social changes and the challenges of managing their future care also make the transition phase a period of major stress and unfavorable consequences [10,11]. CCSs who are going through the transition period may experience an interruption in their medical follow-up, the emergence of medical complications, or the deterioration of their general health [12]. These major challenges highlight the need for supportive strategies as CCSs transition through adolescence and enter young adulthood [13,14].

All existing reviews focused on narrow topics, e.g., interventions for psychosocial and behavioral symptoms only [15], or summarized care plans and models of improving long-term survival for CCSs [16], and thus do not reflect the complex nature of transition interventions for CCSs. Transition interventions can be used to educate and empower CCSs preparing for transition [8] toward assuming adult roles and functions, which can have a sustainable impact on disease prevention and management in long-term survival [17,18]. Recent guidelines suggest a role for transition interventions to promote health for CCSs [19]. Otth et al. [20,21] conducted a systematic review of the concept of transition, related theories, and stakeholder perspectives, and their findings showed that knowledge and education are key facilitators of transition, indicating that there is an increased demand for interventions to improve the transition for CCSs [6,19]. They suggested that transition care should help them feel more prepared and less overwhelmed with the shift into survivorship [6,10,22]. However, whether eligible patients are receiving theory-based transition interventions that comply with the guidelines remains unknown [23,24], because no study has summarized the features of transition interventions for CCSs.

Schwartz et al. [17] developed a socioecological model of adolescent and young adult readiness for transition (SMART). This model identifies potential targets of intervention for future transition research in medical settings, including knowledge related to disease history; health status/needs and the benefits of transition; skills/self-efficacy related to managing personal health and transition; beliefs related to the transition process or adult care; goals of the transition process; relationships among patients, parents, and providers (pediatric and adult providers); and psychological conditions and emotions related to the transition process (Figure 1).

Therefore, with guidance from SMART, this scoping review examined and evaluated the current literature on programs and interventions designed to support the transition from pediatric- to adult-centered care for CCSs. This information is vital for establishing evidence-based guidelines and directing future research in this area.

## 2. Materials and Methods

### 2.1. Study Design

The reporting in this article followed the Preferred Reporting Items for Systematic Reviews and Meta-Analyses extension for Scoping Reviews (PRISMA-ScR) [25]. The protocol was registered prospectively with the Open Science Framework (Registration DOI: (https://doi.org/10.17605/OSF.IO/VKSFW (accessed on 27 December 2021)).

### 2.2. Search Strategy

The comprehensive literature searches were conducted in PubMed, Web of Science, EMBASE, PsycINFO, CINAHL, Ovid, and Cochrane Library, covering the years from January 2005 to October 2022 and updated on 16 January 2023. This starting date coincided with the publication of the seminal Institute of Medicine report in 2005 called “From Cancer Patient to Cancer Survivor: Lost in Transition,” which highlighted cancer survivors and the concept of transition [7,10].

All identified keywords and index terms in a preliminary search strategy were then undertaken across all included databases (Appendix A). The reference list of all identified reports and articles was also searched for additional studies. The final search strategy for Ovid can be found in Appendix A.

### 2.3. Data Management and Study Selection

All retrieved articles were imported into EndNote (version 20.0) and deduplicated. The titles and abstracts of the retrieved articles were independently reviewed by three trained reviewers (MJ, XXL, and ZSQ) based on the eligibility criteria. The full texts were independently reviewed by the same three trained reviewers for inclusion. Reviewers met regularly to discuss progress, and discrepancies were resolved through consensus. Figure 1 presents the study selection process.

### 2.4. Eligibility Criteria

Population: the underlying purpose of this review was to identify intervention programs designed to facilitate the transition to adulthood for CCSs, and these studies may include populations including adolescents and young adults [26], so we did not limit the specific ages of included study subjects.

Concept: CCSs who receive treatment and follow-up in a children’s hospital eventually need to transfer to an adult-oriented center for long-term follow-up care as adults, which has been defined as “transition” [14,27]. The interventions of interest were any discrete transition interventions (any design) that aimed to facilitate the transition to adulthood for CCSs and fulfill at least one of the potential intervention targets indicated in SMART: knowledge, skills/efficacy, belief/expectations, goals, relationships, and psychosocial functioning (Figure 1) [25].

Context: The interventions must include some specific content for the transition of CCSs. No limitations were set on the type of intervention or its duration, the cultural/sub-cultural factors, geographic location, specific racial or gender-based interests, or details about the specific setting.

Study designs: peer-reviewed full-text manuscripts and development or protocol papers were included.

### 2.5. Data Extraction and Quality Appraisal

Data extraction was conducted to capture the information of interest, including sample characteristics, study design, study outcome (including if it was the stated primary outcome), and primary results of the study.

The publications were categorized into types of evidence: (1) development studies (i.e., those providing program descriptions but no evaluation data); (2) case series; (3) single-arm, pre-post studies; (4) pilot studies; and (5) randomized controlled trials (RCTs). The intervention/program components, the order of development procedure/process of the intervention/program (if applicable), the target population, the setting where the program or intervention took place, any outcomes, and the pertinent results of the interventions were described.

## 3. Results

### 3.1. Included Studies

A total of 2564 articles were retrieved from the database searches, and 1234 duplicates were removed, leaving 1330 articles to screen. The process is summarized in Figure 2. Then, 1267 were removed after applying the inclusion criteria, and the full texts of the remaining 63 articles were screened. Finally, nine studies met the inclusion criteria: four pilot studies [11,28,29,30], three single-arm pre-post studies [31,32,33], one case series [34], and one developmental study [35]. Six trials were undertaken in the USA [28,29,31,33,34,35], one in Canada [11], one in the Netherlands [32], and one in Germany [30]. The features of these nine studies are summarized in Table 1. Although this review did not focus on outcomes, as most of the included studies were pilot studies, brief descriptions of the outcomes are provided in the table.

### 3.2. Age Range of Participants and Timing of Intervention

As shown in Table 1, the participants in nine studies (*N* = 827) were adolescents or young adults who had been diagnosed with cancer in childhood and whose current age ranged from 15 to 29 years. Most studies included participants with leukemia, lymphomas, or solid tumors [28,29,31,32], with leukemia being the most common diagnosis. Roux et al. [34] focused on adult patients harboring brain tumors during childhood or adolescence, and three studies did not report the tumor types [11,30,33]. All studies excluded patients who had previously attended any specialized transitional program. Only three studies included family members of CCSs [30,33,35].

Regarding the timing of intervention, four studies reported that the eligible CCSs in the transition interventions were 0–5 years post-completion of cancer-directed treatment [28,29,31,32], and three studies included CCSs over 15 years [30] or 18 years [11,34] of age.

### 3.3. Providers and Types of Intervention Implementations

The providers and types of implementation for each transition intervention varied greatly. In two studies, individualized transition consultations were led by a pediatric nurse practitioner or a physician [29,34].

The responsibilities of the nurse included contacting patients about the transition process, obtaining informed consent, setting up the patients’ appointments, and having individualized consultations with patients who transitioned to adult care [29]. In another study, transitional consultation was held jointly by pediatric and adult neurosurgeons [34]. Kock et al. [30] developed a mobile application, and Blaauwbroek et al. developed a web-based survivor care plan to provide more flexible transition care for CCSs, in which they could set up a user profile to retrieve individualized transition recommendations. Viola et al. [35] integrated the web and peer mentors to implement the intervention; the mentor–participant pairs were introduced through secure text messages, and then the mentors were responsible for initiating six video calls with their mentees to discuss and apply the content of five web-based modules. In another three studies, the intervention was conducted using a hard-copy transitional workbook [11,28,31]. In Bingen and Kupst’s educational program, clinical and research experts with backgrounds in multiple disciplines were invited to present at one of four sessions of a traditional speaker series and were responsible for delivering survivorship education to CCSs based on identified transition topics [33].

### 3.4. Contents of Interventions

The nine current transition interventions fail to incorporate conceptual frameworks. The overarching principles of SMART for grouping the content of interventions were applied in this scoping review: knowledge, skills/efficacy, beliefs/expectations, goals, relationships, and psychosocial functioning (Figure 1). An overview and summary of the various interventions in terms of the categories and subcategories are provided in Table 2.

#### 3.4.1. Knowledge

The results indicated an increased effort to educate CCSs using different aspects of transition knowledge. Knowledge of cancer and treatment history was highlighted in all seven studies [11,28,29,30,31,32,35], followed by knowledge of the risk of the late effects of cancer [28,29,30,32,33,35]. Almost all participants reported becoming more knowledgeable from the interventions [28,29,31,32], especially with regard to knowledge and awareness of late effects [31,32]. Some participants also reported that the medical history and provider information were the most helpful sections of the intervention [28], in addition to supporting efforts to improve transition, and they felt positive about the intervention [11].

Other types of information highlighted in the interventions varied across studies, including the nature and importance of transitioning [11,33], scheduling of planned follow-up care for CCSs [29,30,32,34], information about adult healthcare providers from transition consultations or worksheets [11,28,34], health promotion messages [11,28,31,32,33,35], and nutritional messages [28,29]. However, because of a lack of detailed descriptions of the transition interventions, the information was not sufficient to separate the intervention components, and the health promotion and nutritional messages could not be linked to related outcomes [11,32].

#### 3.4.2. Skills

Four studies focused on cultivating different skills in CCSs [28,30,32,35]. Kock et al. [30] developed a mobile application, and Blaauwbroek et al. [32,35] developed a web-based transition intervention to improve the management capability of coordinating planned follow-up care. CCSs could arrange their own appointments with healthcare providers or family doctors and set reminders for recommended future examinations [30,32]. Bashore et al. [28] also provided several practical tips for CCSs to cultivate their life skills, such as how to budget money, balance a checkbook, and prepare meals.

#### 3.4.3. Psychosocial Functioning

Some transition interventions targeted the psychosocial functioning of CCSs. Four studies provided strategies for emotional adjustment and difficulty coping among teens and young adults following cancer treatment [28,29,33,35]. In the study of Bashore and Bender [28], the CCSs who completed the study reported decreased general worry. In the study by Shea and colleagues [29], the CCSs reported that the transition visit helped them feel more secure and addressed their emotional symptoms. In studies that did not involve the psychosocial component, the participants pointed out the need to add a section on helping them manage feelings of guilt and grief and mental health challenges and ensure that they can meet their responsibilities [11,32].

#### 3.4.4. Goals and Beliefs

Goals and beliefs are two closely related variables in SMART; goals are related to specific objectives that CCSs hope to achieve through transition, including in the knowledge, skills, psychosocial, academic, and vocational dimensions, and beliefs are related to being confident in achieving the goals [17]. Only three studies addressed educational and vocational goals in the intervention process, such as whether to attend college or not, the kind of career to pursue, and the corresponding necessary steps to take [28,29,33], and one study mentioned that peer mentors could guide the patients during the intervention to reshape or strengthen their motivation and confidence to assume responsibility for their care [33]. None of the included studies mentioned helping survivors establish transition goals in other dimensions or assessing their beliefs during the intervention.

#### 3.4.5. Relationships

Transitioning is not an isolated process, and a collaborative relationship between the child, their parents, providers, and other stakeholders is an important factor for a successful transition [17]. However, among the included studies, only one discussed supportive ways to include family members, the challenges of having parents who do not relinquish control and provided strategies to help build collaborative relationships with peer mentors [35].

## 4. Discussion

### 4.1. Main Findings

This scoping review was conducted to understand the features of transition interventions for CCSs. After the literature was reviewed, data from nine eligible papers were assessed, including four pilot studies [11,28,29,30] and three evaluation studies [31,32,34]. The SMART model includes various constructs for developing and instituting transition programs [17] and provides overarching principles that were used to group the contents of interventions for this review.

This review indicates that there is an increased effort to educate CCSs with transition knowledge, especially information related to cancer and follow-up care. CCSs were reported to be better informed about their medical history as well as educated and motivated to pursue appropriate follow-up care and had turned from passive to active in the transition process [11,28,29,30,31,32,34]. This finding is in line with previous guidelines and SMART; that is, treatment history and late effects have been major components in transition education programs [17,19,36]. However, some participants still reported a lack of information on why they needed transition care and what to expect when transitioning to adult-oriented care [28]. As early as 2011, Bashore et al. reported that CCSs said that they did not learn enough about the event of transition ahead of time [37,38]; yet, several years later, only one study reported information being provided to CCSs about the transition process [11], suggesting the need for more research to focus on this issue [9,37].

This review found that researchers have begun to study the development of the independent skills of survivors in existing transition interventions [28,30,32]. Although the number of studies is limited, this finding is consistent with the guidelines; that is, one of the most important elements is that CCSs need to be trained and empowered to become qualified partners in their own care [39,40,41]. SMART identifies skills/self-efficacy related to managing one’s personal health and transition [17]. CCSs expressed a strong desire to cultivate skills to manage their healthcare [42], hoping to be able to handle a lot of information and have consultations with their providers by themselves [9]. All parents in the studies acknowledged the importance of CCSs establishing independence with regard to their psychological, social, and resource needs [43,44] and cultivating coping skills for likely post-treatment challenges [45]. Some providers even recommended developing a transition class or curriculum that could teach key skills to CCSs and their families [44].

This review found that psychosocial functioning is an essential element of transition programs [28,29], in line with the care needs of CCSs [46,47] and the intervention targets mentioned in SMART [17]. Transitioning is a psychological process of adapting to disruption or change and includes anxiety, uncertainty, and concern over unmet medical needs [37]. This negative psychosocial functioning is the major cause of transition failure [48,49,50]. Therefore, providing psychological education and anticipatory guidance for survivors and caregivers, motivating them to pursue appropriate follow-up care, and helping them become more active in the transition process should be among the priorities in future transition programs [9,48].

Although the essential components were identified in existing interventions, the transition was not highlighted as an active, goal-oriented, collaborative process as emphasized in existing guidelines and theories [14,17]. The PanCare guidelines suggest that transition interventions should consider the medical, psychosocial, educational, and vocational goals of survivors [14]. Schwartz et al. [17] reported that transition should be associated with positive beliefs about transition goals. Sadak et al. [40] identified goal setting as a prerequisite for the successful transition of CCSs, which will allow them to remain diligent in maintaining their health [47]. In addition, SMART describes the transition as a coordinated process, and CCSs, caregivers, and providers in medical settings should form a collaborative community around receiving interventions to improve the process [17]. Caregivers should adjust their caregiving role and find a balance between their involvement and their children’s independence. Furthermore, healthcare providers could help navigate the transition process, which could effectively optimize the self-efficacy of CCSs and their ability to assume appropriate healthcare responsibility [40,51,52]. Therefore, future intervention studies should focus on the coordinated role of the three variables of goals, beliefs, and relationships, which may be key aspects of building bridges across the components of intervention and achieving successful transition [48]. CCSs should set transition goals related to their knowledge, skills, and psychosocial functions; formulate positive beliefs about transition care; and ultimately, successfully transfer to adult care within a supportive network of stakeholders [17,53].

### 4.2. Strengths

A systematic process based on the scoping review guidelines was undertaken to capture and map a broad body of the literature. This review applied SMART, a need- and context-based theoretical framework, to gain insights into intervention characteristics. By extracting the intervention components from the included studies that matched the theoretical framework and classifying them, this review provides important insights into the key components that should be considered in the design of transition interventions but have often been overlooked in more traditional evidence syntheses. Moreover, the contents of transition interventions were summarized in an easily accessible format, and they could potentially be used to aid researchers and program designers in the development and design of future interventions for CCSs.

### 4.3. Limitations

First, due to the large number of databases, we may have overlooked some databases that could provide useful information. Second, consistent with the scoping study methodology, a formal appraisal of study quality was not undertaken. Third, the included studies varied widely, and most of them were not designed to assess efficacy (such as feasibility and pilot studies). As such, no definitive conclusions could be made with regard to the efficacy of the interventions. Researchers need to consider this factor when interpreting or applying the results of this review to areas with other multicultural populations.

In addition, the majority of the included studies were conducted in the USA. There may be significant differences among cultures, the expression of demands, and transition procedures in healthcare systems between countries [54]; understanding what these are and explaining them is challenging in this study. Future intervention studies should aim to address these issues to better improve the transition process and enhance the quality of survival for CCSs.

## 5. Conclusions

The studies reviewed in the current paper identified essential elements for a transition program, such as delivering knowledge, developing skills for coordination of care, and addressing psychosocial needs. This review can aid researchers and program designers in the development and design of future transition interventions targeting CCSs. However, the current transition interventions are generally in their infancy, and major deficits were found, including poorly reported intervention components and procedures, a limited amount of relevant validated outcomes, and failure to incorporate conceptual frameworks and international consensus statements. More high-quality, adequately powered, and well-reported RCTs are needed for the enrichment and standardization of future transition interventions.

## Figures and Tables

**Figure 1 cancers-16-00272-f001:**
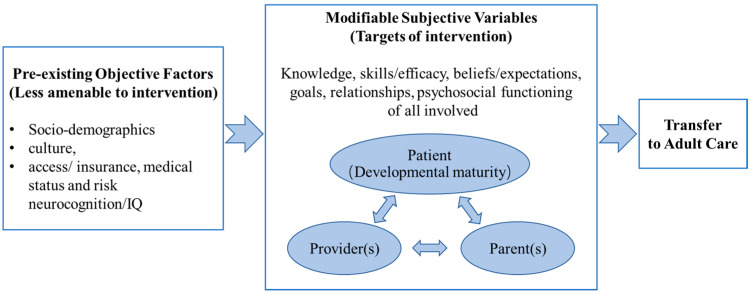
Socioecological model of adolescents’ and young adults’ readiness for transition (Schwartz et al., 2011 [17]).

**Figure 2 cancers-16-00272-f002:**
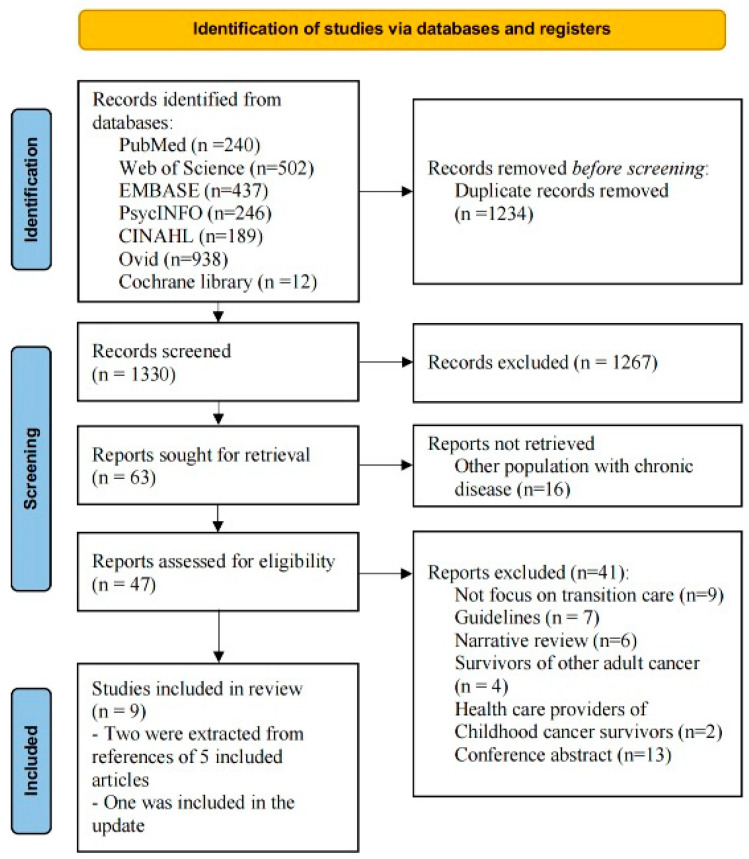
PRISMA flow diagram.

**Table 1 cancers-16-00272-t001:** The features of the included studies.

Author and Year	Origin	Study Design	Population and Sample Size	Conceptual Framework/Theory	Intervention	Intervention Timing	Outcomes
Bingen and Kupst, 2010 [33]	USA	Single-arm, pre-post studies	(1) CCSs (*N* = 99) (2) Caregivers of childhood cancer survivors (*N* = 98) (3) Healthcare providers (*N* = 35) (4) Others (*N* = 26)	None	Educational program	Not mentioned	Accessibility and satisfaction with program
Blaauwbroek, 2012 [32]	The Netherlands	Single-arm, pre-post studies	(1) CCSs (*N* = 80): Leukemia (*n* = 31); malignant lymphoma (*n* = 8); bone tumor (*n* = 13); soft-tissue sarcoma (*n* = 3); Wilms’ tumor (*n* = 7); Langerhans cell histiocytosis (*n* = 7); and others (*n* = 11) (2) Family doctors (*N* = 79)	None	Web-based, survivor care plan driven by family doctor	At least 5 years post-completion of cancer-directed treatment	Accessibility, user-friendliness, and satisfaction with provided information
Bashore, 2016 [28]	USA	Pilot study	CCSs (*N* = 32): Acute lymphoblastic leukemia (*n* = 10); lymphomas (*n* = 4); bone tumors (*n* = 3); Wilms’ tumor (*n* = 2); rhabdomyosarcoma (*n* = 2); and others (*n* = 9)	None	Transition workbook	At least 2 years post-completion of cancer-directed treatment	Worry and transition readiness
Landier, 2015 [31]	USA	Single-arm, pre-post studies	CCSs (*N* = 369): Leukemia (*n* = 165); lymphoma (*n* = 90); and solid tumor (*n* = 114)	None	Tailored education consultation provided by a pediatric nurse practitioner or physician	At least 2 years post-completion of cancer-directed treatment	Awareness of health risks
Kock, 2015 [30]	USA	Pilot study	(1) CCSs (*N* = 13) (2) Accompanying relatives (*N* = 9)	None	Personalized aftercare mobile application	Over 15 years of age	Usability of mobile application
Shea, 2019 [29]	USA	Pilot study	CCSs (*N* = 19): Acute lymphoblastic leukemia (*n* = 12); Hodgkin’s lymphoma (*n* = 3); Langerhans cell histiocytosis (*n* = 2); Burkitt lymphoma (*n* = 1); and mature B-cell lymphoma (*n* = 1)		One-time structured transition visit provided by a pediatric research nurse practitioner	0–6 months post-completion of cancer-directed treatment	Experience with transition visit
Ryan, 2020 [11]	Canada	Pilot study	CCSs (*N* = 16)	None	Educational workbook	Over 18 years of age	Understandability, actionability, and acceptability,
Roux, 2021 [34]	USA	Case series	CCSs (*N* = 14) with brain tumor	None	Joint mixed transitional consultation provided by pediatric and adult neurosurgeons	Over 18 years of age	Benefit
Viola, 2022 [35]	USA	development study	(1) CCSs (*N* = 25) Blood cancer (*n* = 20) Solid tumor (*n* = 4) Brain tumor (*n* = 1) (2) Caregivers of childhood cancer survivors (*N* = 1) (3) Health care providers (*N* = 3)	None	Self-management and Peer-Mentoring Intervention	Not mentioned	Transition readiness

Note: CCS: childhood cancer survivors.

**Table 2 cancers-16-00272-t002:** Intervention elements in existing transition studies.

Categories	Subcategories
Knowledge	Knowledge of cancer and treatment history [11,28,29,30,31,32,34,35]
	Risk for late effects of cancer [28,29,30,32,33,35]
	Scheduling for planned follow-up care and recommended screenings [29,30,32,34]
	Information on healthcare providers involved in transition care [11,28,34]
	Introduction, description of the transition process, and information on the importance of transition [11,33]
	Health promotion messages [11,28,31,32,33,35]
	Nutrition [28,29]
Skills	Enhancement of the management capabilities of coordinating planned follow-up care and recommended screenings [30,32,35]
	Cultivation of life skills: how to budget money, balance a checkbook, and prepare meals [28]
Psychosocial functioning	Addressing mental health needs and providing support services [28,29,33,35]
Goals	Setting up educational and vocational goals: whether they wanted to attend college and how they envisioned this happening [28,29,35]
Belief	Review motivation and confidence to assume responsibility for care [35]
Relationships	Discuss the supportive ways of including family members and the challenges of parents who do not relinquish control [35]

## Data Availability

The data can be shared up on request.

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
