# Peer review of "Features of Studies on Transition Interventions for Childhood Cancer Survivors: A Scoping Review"

_cancers, 2024, doi:10.3390/cancers16020272_

Round 1
Reviewer 1 Report
Comments and Suggestions for Authors
Ma Jun et Colleagues provided an extensive review of the literature about an interesting field. The manuscript is methodologically appropriate and materials well organized.
Nevertheless, before acceptance of the manuscript some minor aspects should be implemented:
1- Page 2, line 69-71. The sentence is a methodological consideration that didn’t add essential information to the topic. Consider to remove it.
2- Page 3, Line 113. Please move the consideration about inclusion criteria in the specific section (2.4).
3- Page 3, Line 121. The first sentence of the paragraph about inclusion criteria seems to be confused. I suggest to rephrase it.
4- Page 4, Line 161. Please add the number of the Table.
5- Figure 2. In the box of included studies, please check the word “exacted” and remove a bracket.
6- Page 5, Line 166. The mean range of 15-29 years refers to the diagnosis or the time of the studies? Please, specify.
7- Page 7, Line 178. The word “delivery” to identify interventions and their modality for survivors is not so commonly use. Please, consider if there is a better synonym.
8- Table 2. Please, consider to use a more informative title.
Comments on the Quality of English Language
The manuscript needs a revision by a native English speaker.
Author Response
Dear Ms. Sierra Peng and reviewers,
Thank you for reviewing our manuscript (ID cancers-2726562, entitled " Features of studies on transition interventions for childhood cancer survivors: A scoping review). We found the comments very helpful in improving our manuscript. Below we list the reviewer comments and our responses. We had presented the updated changes in “italic”.
Reviewer 1:
comments and Suggestions for Authors
Ma Jun et Colleagues provided an extensive review of the literature about an interesting field. The manuscript is methodologically appropriate and materials well organized.
Nevertheless, before acceptance of the manuscript some minor aspects should be implemented:
- Page 2, line 69-71. The sentence is a methodological consideration that didn’t add essential information to the topic. Consider to remove it.
Response: Thank you very much, we have removed that part.
- Page 3, Line 113. Please move the consideration about inclusion criteria in the specific section (2.4).
Response: Thank you very much for your suggestions, we have made changes here.
- Page 3, Line 121. The first sentence of the paragraph about inclusion criteria seems to be confused. I suggest to rephrase it.
Response: Thanks to your suggestion, we have reorganized the definition of transition and hopefully made it clearer.
- Page 4, Line 161. Please add the number of the Table.
Response: Thanks for the suggestion, have added the table where it fits!
- Figure 2. In the box of included studies, please check the word “exacted” and remove a bracket.
Response: Thank you very much, we have revised figure 2.
- Page 5, Line 166. The mean range of 15-29 years refers to the diagnosis or the time of the studies? Please, specify.
Response: Thank you for the suggestion, this refers to the age of participation in these studies, not the age of diagnosis, and we have reworded the sentence to make it clearer.
- Page 7, Line 178. The word “delivery” to identify interventions and their modality for survivors is not so commonly use. Please, consider if there is a better synonym.
Response: Thanks to your suggestion, we have replaced delivery with "Intervention implementation", and hopefully can make it clearer.
- Table 2. Please, consider to use a more informative title.
Response: Thanks, we have modified the title of table 2 to make it more relevant to the content therein.
- The manuscript needs a revision by a native English speaker.
Response: Thanks, we used the editing services listed at
https://www.mdpi.com/authors/english.
Again, thank you editor and reviewers. We are grateful for your consideration and for the potential opportunity to publish this paper in the Cancers.
Honghong Wang

Reviewer 2 Report
Comments and Suggestions for Authors
Thanks for the Scoping Review of an important topic in childhood cancer survivorship.
I had a couple of general questions about the terminology used in this paper. You mention you conducted a systematic scoping review, which may be confusing to some readers. Maybe a better way to state this is that you used a systematic approach to completing the scoping review of the evidence. A systematic review is markedly different from a scoping review due to the strict nature of the systematic review inclusion/exclusion criteria.
Your methodology is sound and well done. You focused only the on the transition process and the interventions as stated in the objectives.
Your results are well presented and the discussion is clear and comprehensive. You made clear recommendations as noted in the evidence that psychological/emotional functioning are critical components of a potentially successful transition plan.
Author Response
Dear Ms. Sierra Peng and reviewers,
Thank you for reviewing our manuscript (ID cancers-2726562, entitled " Features of studies on transition interventions for childhood cancer survivors: A scoping review). We found the comments very helpful in improving our manuscript. Below we list the reviewer comments and our responses. We had presented the updated changes in “italic”.
* * * * *
Reviewer 2:
Thanks for the Scoping Review of an important topic in childhood cancer survivorship.
- I had a couple of general questions about the terminology used in this paper. You mention you conducted a systematic scoping review, which may be confusing to some readers. Maybe a better way to state this is that you used a systematic approach to completing the scoping review of the evidence. A systematic review is markedly different from a scoping review due to the strict nature of the systematic review inclusion/exclusion criteria.
Response: Thanks to your suggestion, we have changed the wording throughout the text to differentiate between scoping reviews and systematic review.
- Your methodology is sound and well done. You focused only the on the transition process and the interventions as stated in the objectives.
Response: Greatly appreciated! because the concept, theory, and cross-sectional studies of transition have been stated in previous reviews, and the main purpose of this paper is to focus on transition-related interventions, summarize the current state of intervention features, and provide recommendations for the development of future interventions.
- Your results are well presented and the discussion is clear and comprehensive. You made clear recommendations as noted in the evidence that psychological/emotional functioning are critical components of a potentially successful transition plan.
Response: Many thanks!
Again, thank you editor and reviewers. We are grateful for your consideration and for the potential opportunity to publish this paper in the Cancers.
Honghong Wang

Reviewer 3 Report
Comments and Suggestions for Authors
Overview: This manuscript presents a scoping review on transition interventions for childhood cancer survivors. This is a very important topic and a scoping review is appropriate for the research questions. The general approach to the review is detailed and rigorous, with clear research questions, prospective registration of the protocol, the use of multiple reviewers, and reporting following PRIMSA guidelines.
Major Comments:
1. While the Introduction clearly states the focus of this review on transition, there are other recent reviews that have some overlapping content/questions. For example, Mobley et al 2023, “Interventions to address disparities and barriers to pediatric cancer survivorship care: A scoping review” and Zhang et al. 2021, “Psychosocial, behavioral, and supportive interventions for pediatric, adolescent, and young adult cancer survivors: A systematic review and meta-analysis.” This manuscript would be strengthened if the authors briefly discuss these review papers and add a statement of how this scoping review uniquely contributes to the field.
2. The limitations section notes that most studies were conducted in the USA; this point should be expanded to note how healthcare systems differ between countries and interventions must consider transition in the context of their healthcare systems.
3. The conclusions section notes “failure to incorporate conceptual frameworks” as a major limitation of the existing literature, but conceptual framework is not reported in Table 1 or in the text describing the results. This should be added to results to provide evidence of the conclusion.
Comments on the Quality of English Language
Minor edits needed for grammar/awkward phrasing. For example, on p. 9, lines 284 - rephrase "and have lone consultation with providers"
Author Response
Dear Ms. Sierra Peng and reviewers,
Thank you for reviewing our manuscript (ID cancers-2726562, entitled " Features of studies on transition interventions for childhood cancer survivors: A scoping review). We found the comments very helpful in improving our manuscript. Below we list the reviewer comments and our responses. We had presented the updated changes in “italic”.
* * * * *
Reviewer 3:
Major Comments:
- While the Introduction clearly states the focus of this review on transition, there are other recent reviews that have some overlapping content/questions. For example, Mobley et al 2023, “Interventions to address disparities and barriers to pediatric cancer survivorship care: A scoping review” and Zhang et al. 2021, “Psychosocial, behavioral, and supportive interventions for pediatric, adolescent, and young adult cancer survivors: A systematic review and meta-analysis.” This manuscript would be strengthened if the authors briefly discuss these review papers and add a statement of how this scoping review uniquely contributes to the field.
Response: Many thanks, we have revised this part.
- The limitations section notes that most studies were conducted in the USA; this point should be expanded to note how healthcare systems differ between countries and interventions must consider transition in the context of their healthcare systems.
Response: Many thanks, we have revised this part.
- The conclusions section notes “failure to incorporate conceptual frameworks” as a major limitation of the existing literature, but conceptual framework is not reported in Table 1 or in the text describing the results. This should be added to results to provide evidence of the conclusion.
Response: Thanks, we have added to the tables and results section about the conceptual framework.
- Comments on the Quality of English Language
Minor edits needed for grammar/awkward phrasing. For example, on p. 9, lines 284 - rephrase "and have lone consultation with providers"
Response: Thanks, we used the editing services listed at
https://www.mdpi.com/authors/english.
Again, thank you editor and reviewers. We are grateful for your consideration and for the potential opportunity to publish this paper in the Cancers.
Honghong Wang

Reviewer 4 Report
Comments and Suggestions for Authors
Dear Authors,
congratulations on your work on this interesting topic.
As regards the Materials and Methods section, the protocol registered with the Open Science Framework is incomplete and does not include all relevant data (lines 89-90). Please replace the link with one that opens the corresponding page directly (https://doi.org/10.17605/OSF.IO/VKSFW).
Line 94: is the consultant librarian among the authors? If not, consider of accrediting her/his work in the Acknowledgments section.
Line 95: Scopus is a large database that was not included in your search strategy. Was there any particular reason?
In my opinion, the Materials and Methods section is too wordy, and this makes the manuscript hard to follow for the readership. The referral to the PRISMA-ScR Checklist could save some space. The omission of unnecessary information is also encouraged (lines 98-100, etc.).
Line 115: According to your statement, there were three reviewers, while in your team’s preceding publication, there were two (10.1136/bmjopen-2023-074162). Why was that change applied?
Line 118: Since included studies were limited in number, how many studies were found with discrepancies that needed to be resolved through consensus?
Line 156: What do you mean by sery? Maybe case series?
Line 161: Please refer to the exact table number.
Table 1: Since you have used the acronym CCS even in the abstract, restrain from referring to them as “childhood cancer survivors” in the Table and use the simplified form where needed.
Lines 255-256: When the same citation is used, and no other citation is between them, then please refer to the citation only once at the end of the sentence.
There is one policy document, namely “Comprehensive Cancer Care for Children and Their Families” (2015), available at https://nap.nationalacademies.org/catalog/21754/comprehensive-cancer-care-for-children-and-their-families-summary-of and could comment on that, because it refers to SMART.
In the Discussion section there are missing some key publications on the field. Here are 5 representative works from my perspective:
(i) Cheng, L., Mao, X., Chen, Q., Pu, H., & Yu, L. (2023). Identifying the Distinct Profiles of Transition Readiness in Chinese Pediatric Cancer Survivors. Cancer nursing, 46(3), 189–197. https://doi.org/10.1097/NCC.0000000000001195
(ii) Casillas, J., Kahn, K. L., Doose, M., Landier, W., Bhatia, S., Hernandez, J., Zeltzer, L. K., & Padres Contra El Cáncer (2010). Transitioning childhood cancer survivors to adult-centered healthcare: insights from parents, adolescent, and young adult survivors. Psycho-oncology, 19(9), 982–990. https://doi.org/10.1002/pon.1650
(iii) Marchak, J. G., Sadak, K. T., Effinger, K. E., Haardörfer, R., Escoffery, C., Kinahan, K. E., Freyer, D. R., Chow, E. J., & Mertens, A. (2023). Transition practices for survivors of childhood cancer: a report from the Children's Oncology Group. Journal of cancer survivorship : research and practice, 17(2), 342–350. https://doi.org/10.1007/s11764-023-01351-y
(iv) Prussien, K. V., Barakat, L. P., Darabos, K., Psihogios, A. M., King-Dowling, S., O'Hagan, B., Tucker, C., Li, Y., Hobbie, W., Ginsberg, J., Szalda, D., Hill-Kayser, C., & Schwartz, L. A. (2022). Sociodemographics, Health Competence, and Transition Readiness Among Adolescent/Young Adult Cancer Survivors. Journal of pediatric psychology, 47(10), 1096–1106. https://doi.org/10.1093/jpepsy/jsac039
(v) Psihogios, A. M., Schwartz, L. A., Deatrick, J. A., Ver Hoeve, E. S., Anderson, L. M., Wartman, E. C., & Szalda, D. (2019). Preferences for cancer survivorship care among adolescents and young adults who experienced healthcare transitions and their parents. Journal of cancer survivorship : research and practice, 13(4), 620–631. https://doi.org/10.1007/s11764-019-00781-x
Are you aware of the dimensions outlined in Levesque's framework? And why have you refrained from citing the corresponding works?
Outcomes from the citation [18] should be elaborated in the text.
A Table with the Summary of Measurement Themes for Abstracted Quality Indicators (from the 9 selected publications) could be built and help the readership on measurement themes (Satisfaction, Self-management/self-efficacy, Clinic attendance, Transition readiness, Medication and treatment adherence, Health status, Quality of life, Autonomy, Disease knowledge, Social development, Health care engagement, Self-advocacy, Resilience, Transition education, Transition planning, Acknowledgment of emerging adult, Primary care involvement, Communication, Coordination of care, Patient/provider relationship, Flexibility of transition, Continuity of care, Transfer documentation, Transfer handover, Acute care utilization, Quality of care, Transition policy, Information technology, Parent/caregiver roles & responsibilities). A modified AGREE II instrument would also be helpful in assessing the identified quality indicators.
Thank you.
Comments on the Quality of English Language
Please double-check the manuscript for typos and make the necessary changes. Acronyms should be used throughout the text and not selectively.
Author Response
Dear Ms. Sierra Peng and reviewers,
Thank you for reviewing our manuscript (ID cancers-2726562, entitled " Features of studies on transition interventions for childhood cancer survivors: A scoping review). We found the comments very helpful in improving our manuscript. Below we list the reviewer comments and our responses. We had presented the updated changes in “italic”.
* * * * *
Reviewer 4:
congratulations on your work on this interesting topic.
- As regards the Materials and Methods section, the protocol registered with the Open Science Framework is incomplete and does not include all relevant data (lines 89-90). Please replace the link with one that opens the corresponding page directly (https://doi.org/10.17605/OSF.IO/VKSFW).
Response: Thank you very much for your suggestion, we have updated the registration information, and provide an activatable link.
- Line 94: is the consultant librarian among the authors? If not, consider of accrediting her/his work in the Acknowledgments section.
Response: Many thanks, we have revised this section.
- Line 95: Scopus is a large database that was not included in your search strategy. Was there any particular reason?
Response: Many thanks! There is no particular reason, this was an oversight on our part, which we have stated in the limitation and will improve in future studies.
- In my opinion, the Materials and Methods section is too wordy, and this makes the manuscript hard to follow for the readership. The referral to the PRISMA-ScR Checklist could save some space. The omission of unnecessary information is also encouraged (lines 98-100, etc.).
Response: Thanks for the suggestion, we've made changes to the section to hopefully make it look cleaner!
- Line 115: According to your statement, there were three reviewers, while in your team’s preceding publication, there were two (10.1136/bmjopen-2023-074162). Why was that change applied?
Response: Thank you for your question, yes these are our team members, in another article we used a realist synthesis to develop a conceptual framework, which is a different study than this article, hence we enrolled the different key members.
- Line 118: Since included studies were limited in number, how many studies were found with discrepancies that needed to be resolved through consensus?
Response: Thank you for your suggestion, this was one of the steps in our literature screening process, where the citation [33] was discussed to resolve disagreements, and it was ultimately felt that despite being different from the usual intervention study design, the literature still contained elements of the intervention and so the final decision was to include it.
- Line 156: What do you mean by sery? Maybe case series?
Response: Thank you for your question, but we did not find the "sery" you mentioned in line156, we mentioned the category "case series" in line141.
- Line 161: Please refer to the exact table number.
Response: Thanks, we have added it.
- Table 1: Since you have used the acronym CCS even in the abstract, restrain from referring to them as “childhood cancer survivors” in the Table and use the simplified form where needed.
Response: Thanks for the suggestion, we've revised it
- Lines 255-256: When the same citation is used, and no other citation is between them, then please refer to the citation only once at the end of the sentence.
Response: Thanks for the suggestion, we've revised it.
- There is one policy document, namely “Comprehensive Cancer Care for Children and Their Families” (2015), available at https://nap.nationalacademies.org/catalog/21754/comprehensive-cancer-care-for-children-and-their-families-summary-of and could comment on that, because it refers to SMART.
Response: Thank you for your suggestion, this framework be used for guiding information extraction in this study, and we will also carefully study this conference summary you mentioned, and apply useful insights to our upcoming development of an intervention program.
- In the Discussion section there are missing some key publications on the field. Here are 5 representative works from my perspective:
(i) Cheng, L., Mao, X., Chen, Q., Pu, H., & Yu, L. (2023). Identifying the Distinct Profiles of Transition Readiness in Chinese Pediatric Cancer Survivors. Cancer nursing, 46(3), 189–197. https://doi.org/10.1097/NCC.0000000000001195IF: 2.760 Q1
(ii) Casillas, J., Kahn, K. L., Doose, M., Landier, W., Bhatia, S., Hernandez, J., Zeltzer, L. K., & Padres Contra El Cáncer (2010). Transitioning childhood cancer survivors to adult-centered healthcare: insights from parents, adolescent, and young adult survivors. Psycho-oncology, 19(9), 982–990. https://doi.org/10.1002/pon.1650IF: 3.955 Q2
(iii) Marchak, J. G., Sadak, K. T., Effinger, K. E., Haardörfer, R., Escoffery, C., Kinahan, K. E., Freyer, D. R., Chow, E. J., & Mertens, A. (2023). Transition practices for survivors of childhood cancer: a report from the Children's Oncology Group. Journal of cancer survivorship : research and practice, 17(2), 342–350. https://doi.org/10.1007/s11764-023-01351-yIF: 4.062 Q1
(iv) Prussien, K. V., Barakat, L. P., Darabos, K., Psihogios, A. M., King-Dowling, S., O'Hagan, B., Tucker, C., Li, Y., Hobbie, W., Ginsberg, J., Szalda, D., Hill-Kayser, C., & Schwartz, L. A. (2022). Sociodemographics, Health Competence, and Transition Readiness Among Adolescent/Young Adult Cancer Survivors. Journal of pediatric psychology, 47(10), 1096–1106. https://doi.org/10.1093/jpepsy/jsac039IF: 3.624 Q2
(v) Psihogios, A. M., Schwartz, L. A., Deatrick, J. A., Ver Hoeve, E. S., Anderson, L. M., Wartman, E. C., & Szalda, D. (2019). Preferences for cancer survivorship care among adolescents and young adults who experienced healthcare transitions and their parents. Journal of cancer survivorship : research and practice, 13(4), 620–631. https://doi.org/10.1007/s11764-019-00781-xIF: 4.062 Q1
Response: Thank you for the suggestion, scholars have conducted a certain number of qualitative studies of transitions as well as cross-sectional studies, which provide evidences for developing intervention programs, and we have added some of this literature that you mentioned.
- Are you aware of the dimensions outlined in Levesque's framework? And why have you refrained from citing the corresponding works?
Response: Thank you for your suggestion, we read carefully the literature "Levesque JF, Harris MF, Russell G. Patient-centred access to health care: conceptualizing access at the interface of health systems and populations. Int J Equity Health. 2013;12:18. Published 2013 Mar 11. doi:10.1186/1475-9276-12-18". This framework described the supply and demand side factors of access to healthcare resources. The framework contains five dimensions of accessibility that affect access to healthcare resources: 1) Approachability; 2) Acceptability; 3) Availability and accommodation; 4) Affordability; and 5) Appropriateness. It does not seem relevant to our study so we did not cite it.
- Outcomes from the citation [18] should be elaborated in the text.
Response: Thanks for the suggestion, we've revised it.
Again, thank you editor and reviewers. We are grateful for your consideration and for the potential opportunity to publish this paper in the Cancers.
Honghong Wang

Round 2
Reviewer 4 Report
Comments and Suggestions for Authors
Dear Authors,
Thank you for the revision of the manuscript, and I hope that you feel that this process is in favor of the readership.
The typo (I guess) in line 156 of the first version ("one case sery [33]") has been replaced with "one case study [35]" in line 139 of the second version. So, you probably found it and corrected it according to another reviewer's comments.
Access and insurance are pre-existing objective factors in the SMART model (as demonstrated in Figure 1 of your manuscript), so the dimensions outlined in Levesque's framework are not that irrelevant.
I have not received any response to my comment on building a Table with the Summary of Measurement Themes for Abstracted Quality Indicators from the 9 selected publications.
Thank you
Comments on the Quality of English Language
The manuscript has been extensively edited in terms of the English language, but there are still some minor mistakes (e.g., Line 46, "CCSs who experience the transition period may experience...", etc.). The certificate that is provided confirms that the manuscript will be readable if published.
Author Response
Dear reviewer,
Thank you for reviewing our manuscript (ID cancers-2726562, entitled " Features of studies on transition interventions for childhood cancer survivors: A scoping review). We found the comments very helpful in improving our manuscript. Below we list the your comments and our responses. We had highlighted the updated changes in “green”.
* * * * *
Reviewer 4:
- The typo (I guess) in line 156 of the first version ("one case sery [33]") has been replaced with "one case study [35]" in line 139 of the second version. So, you probably found it and corrected it according to another reviewer's comments.
Response: many thanks.
- Access and insurance are pre-existing objective factors in the SMART model (as demonstrated in Figure 1 of your manuscript), so the dimensions outlined in Levesque's framework are not that irrelevant.
Response: many thanks. This framework described the supply and demand side factors of access to healthcare resources, some of which are pre-existing objective factors in the SMART model. In this review, however, we primarily summarize what is relevant to the intervention process and strategies, using the second part of SMART-modifiable subjective variables as reference. In another ongoing realist synthesis study we published in BMJ OPEN that you mentioned earlier, we systematically summarized the pre-existing demographic, disease and treatment factors; contextual, mechanisms, and outcome variables that influence transition process, where we will use this framework as an important reference. Thanks again for your kind suggestions.
- A Table with the Summary of Measurement Themes for Abstracted Quality Indicators (from the 9 selected publications) could be built and help the readership on measurement themes (Satisfaction, Self-management/self-efficacy, Clinic attendance, Transition readiness, Medication and treatment adherence, Health status, Quality of life, Autonomy, Disease knowledge, Social development, Health care engagement, Self-advocacy, Resilience, Transition education, Transition planning, Acknowledgment of emerging adult, Primary care involvement, Communication, Coordination of care, Patient/provider relationship, Flexibility of transition, Continuity of care, Transfer documentation, Transfer handover, Acute care utilization, Quality of care, Transition policy, Information technology, Parent/caregiver roles & responsibilities). A modified AGREE II instrument would also be helpful in assessing the identified quality indicators.
Response: many thanks. We apologize for missing this important suggestion. The main reason we have not provided this table: As we stated in lines 143-145, existing intervention studies are still in infancy, and the quality indicators are more limited to feasibility and acceptability, etc. Therefore, this review did not focus on these measurement themes, we just provided a brief descriptions of measurement themes in table 1. Therefore, these important measurement themes you mention are not summarized in this paper. However, in the other ongoing realist synthesis study you mentioned earlier, we have systematically summarized the short-term and long-term quality evaluation indicators of transition process, which we will present in a future paper. Thank you again for your advice and patience.
- Comments on the Quality of English Language: The manuscript has been extensively edited in terms of the English language, but there are still some minor mistakes (e.g., Line 46, "CCSs who experiencethe transition period may experience...", etc.). The certificate that is provided confirms that the manuscript will be readable if published.
Response: many thanks. We again carefully scrutinized the language throughout the text.
Again, thank you editor and reviewers. We are grateful for your consideration and for the potential opportunity to publish this paper in the Cancers.
Honghong Wang
